# Absorption/Attenuation Spectral Description of ESKAPEE Bacteria: Application to Seeder-Free Culture Monitoring, Mammalian T-Cell and Bacteria Mixture Analysis and Contamination Description

**DOI:** 10.3390/s23094325

**Published:** 2023-04-27

**Authors:** Bruno Wacogne, Marine Belinger Podevin, Naïs Vaccari, Claudia Koubevi, Céline Codjiová, Emilie Gutierrez, Pauline Bourgeois, Lucie Davoine, Marjorie Robert-Nicoud, Alain Rouleau, Annie Frelet-Barrand

**Affiliations:** 1Institut FEMTO-ST, Université de Franche-Comté, CNRS, F-25000 Besançon, France; 2Centre d’Investigation Clinique, Centre Hospitalier Universitaire de Besançon, INSERM CIC 1431, 25030 Besançon, France; 3Smaltis, Bioinnovation, Rue Charles Bried, 25030 Besançon, France

**Keywords:** white light spectroscopy, absorption spectra, ESKAPEE bacteria, *Lactococcus lactis*, co-culture monitoring, mammalian T-cells, contamination

## Abstract

Despite numerous innovations, measuring bacteria concentrations on a routine basis is still time consuming and ensuring accurate measurements requires careful handling. Furthermore, it often requires sampling small volumes of bacteria suspensions which might be poorly representative of the real bacteria concentration. In this paper, we propose a spectroscopy measurement method based on a description of the absorption/attenuation spectra of ESKAPEE bacteria. Concentrations were measured with accuracies less than 2%. In addition, mixing the mathematical description of the absorption/attenuation spectra of mammalian T-cells and bacteria allows for the simultaneous measurements of both species’ concentrations. This method allows real-time, sampling-free and seeder-free measurement and can be easily integrated into a closed-system environment.

## 1. Introduction

### 1.1. Context

Nowadays, the rapid and sensitive detection of pathogens is one of the most challenging aspects in the fields of human health and the agri-food industry. It is, therefore, a key factor for the diagnosis of diseases and treatments, and for quality control during the industrial processes of drug/biosimilar production. Numerous techniques have been developed to detect, identify and enumerate bacteria, as reviewed in [1].

Historically, detection techniques have been based on growth monitoring and measuring different changes: (i) gas composition in a closed environment, used for the blood bacteria detection of CO_2_ in sepsis diagnosis [2]; (ii) the production of highly charged ionic metabolites inducing changes in the electrical properties of culture mediums measurable by impedance techniques [3,4]; (iii) ATP as a marker of microorganism viability but requiring filtration to distinguish the bacteria from other sources [5]; (iv) thermal changes measurable by microcalorimetry [6]; and (v) optical density-based techniques measured at specific wavelengths. The sensitivity of each method is dependent on the types of microorganisms. The need for bacterial growth prior to analysis limits these measurements only to use with cultivable bacteria. 

Recently, improvements have been achieved through the use of fluorescence, which allows detection techniques to be more sensitive and accurate, allows for simultaneous detections, are high speed, relatively low cost, do not need the growth of bacteria and have a short detection time [7]. Plate seeders/readers allow researchers to detect and quantify fluorescent *E. coli* in combination with optical density using microspheres [8]. Flow cytometry could also be employed using viability-activated fluorophores. It allows for fast counting while using multiple fluorophores, displaying a high sensitivity and specificity [9]. One derivative of this technique (DEFT direct epifluorescence filtration technique [10]) has been developed for use on the solid phase after trapping by a filter and allows the detection of only the viable bacteria. All these techniques require labeling the microorganisms with fluorophores.

### 1.2. More Innovative Techniques

New innovative detection methods have been developed, allowing for detection without cultures, faster/in real-time detection, the improvement of sensitivity and/or specificity, possible automation, size reduction and the development of cost-effective devices. These methods are based on a recognition of the relationship between ligands and bacteria. Ligands increase sensitivity and allow for earlier detection but still require sampling and/or cultivation. They can be of different types and display a broad spectrum for the detection of any bacteria: (i) antibodies; (ii) aptamers, which have emerged as a good alternative to antibodies since they are cheaper and could target common elements of Gram^−^ or Gram^+^ bacteria [11]; (iii) bacteriophages [12]; (iv) AMPs, which present a strong affinity to bacterial membranes, are stable, easy to produce and store and display a low limit of detection within less than 30 min of incubation [13]; and (v) fluorogenic RNA-cleaving DNAzymes [14], which allow for the detection of 1 bacteria.mL^−1^ in blood [15]. They could be used in combination with optical methods for the label-free detection of bacteria [16] including SPR, QCM and Raman [17], and these techniques could be developed for on-line, real-time measurements and integrated into biosensors. Other techniques based on microfluidics (acoustophoresis, microdroplet) and microscopy (reviewed in [7]) are also used for bacteria detection.

### 1.3. Commercial Systems

Already available for use on mammalian cells, commercial systems to count bacteria have started to emerge. They take advantage of physical properties to help determine bacteria concentrations. Different systems exist which either count bacteria plated via an automatic method (Scan 500 (Interscience), Mica (Diamidex)….), using oxygen-coupled fluorescence compounds (BACTEC (BD) [18]), or directly within suspensions. Bactobox is a portable device developed by SBT Instruments which uses impedance flow cytometry [19]. This label-free technology is based on the principle of impedance creation within fluids after an electric current application, which is distinctive for almost any bacteria (including anaerobic) present within the sample, but can also detect other biological elements which should be removed before taking measurements. Other systems use fluorescence, such as Quantom-tx (Logos Biosystems [20]), an image-based platform that allows for total and viable counts to be obtained directly from the samples without the need for prior cultivation, but it requires sampling and pre-incubation with fluorochromes under specific conditions.

### 1.4. Actual Needs

The classic plate culture approach is still widely used because it is simple and cost-effective; however, one of its major drawbacks is the time required for obtaining measurement results. In addition, all the above-mentioned techniques still require culturing steps prior to analysis to increase the bacteria concentration, and/or for the sampling of a small volume (which breaks the sterility of closed cultivation systems), and/or the addition of compounds and/or fluorophores to increase sensitivity. This highlights the need for rapid, effective, reproducible, cost-effective, label-free, real-time and automatable techniques for bacteria monitoring. At present, no technique fulfills all these criteria.

Within the last few years, white light spectroscopy has proven to be a powerful tool to count mammalian cells. Because measurements are performed on large volume (3 mL), they are highly representative of the suspension content. White light spectroscopy can easily be integrated into a contactless system without sampling for real-time measurements [21,22,23,24].

In this paper, we propose a white light spectroscopy method to enumerate and monitor bacteria cultivation. The technique is demonstrated with bacteria from the ESKAPEE group, a highly virulent and antibiotic-resistant bacteria responsible for hospital-acquired diseases and which includes Gram^+^ and Gram^−^ bacteria. A mathematical description of the shapes of the absorption/attenuation spectra allows for the measurement of bacteria concentrations and for the simultaneous measurement of the two species’ concentrations within cell–bacteria mixtures. Section 2 describes the material and methods used in this work. Section 3 explains how the mathematical description of the absorption/attenuation spectra can be used to measure bacteria concentrations. It also shows examples of the measurement of simultaneous concentrations of mammalian T-cells and bacteria co-cultures, and illustrates how a contaminated T-cell culture can be described. Mammalian T-cells are considered because they are part of studies conducted in the framework of ATMP production (not yet published). Section 4 proposes a discussion of the obtained results. Conclusions will be drawn in Section 5.

## 2. Materials and Methods

The main goal of this work was to determine the absorption/attenuation spectral properties of bacteria belonging to the ESKAPEE group. However, *Lactococcus lactis* was also considered as it is a Gram-positive model bacterium used in some experiments presented here.

### 2.1. Bacteria Preparation

The bacteria, culture media and buffers used in the present study are listed in Table 1 and Table 2, respectively. All the bacteria were stored long term at −80 °C in LB medium/30% glycerol (*v*/*v*) and thawed on a TSA gelose and incubated overnight, except for *L. lactis*, which was stored at −80 °C in M17/15% glycerol and thawed on a M17A medium containing 0.5% glucose for 2 days [25]. Precultures were made with 4 different clones in 10 mL medium (LB, except TSB was used for *E. cloacae*) overnight at 225 rpm, while M17B/0.5% glucose was used for *L. lactis*. Then, the precultures were centrifuged at 7180× *g* for 10 min at RT. Afterwards, the pellets were resuspended in an appropriate volume of PBS to adjust to an optical density of 1 ± 0.05 at 600 nm using a spectrophotometer (Biowave DNA, Biochrom Ltd., Cambridge, UK). The bacterial suspensions (10 mL) were then centrifuged at 9000× *g* for 15 min at RT and the pellets were resuspended in 1 mL of PBS.

### 2.2. Bacteria Enumeration

Suspensions were then diluted following a half dilution in PBS 1X pH 7.4 to obtain a range of concentrations between 1 × 10^8^ and 7.81 × 10^5^ cfu.mL^−1^ (8 tubes) in 4 mL. Theoretical concentrations were determined on the basis that 1 DO unit at 600 nm = 1 × 10^9^ cfu.mL^−1^. An amount of 3 mL from each tube was transferred into spectroscopic cuvettes to perform spectral measurements. The remaining volume was then diluted into 9 mL of NaCl 0.85% prior to appropriate dilutions within the same buffer for enumeration. A volume of 54.3 µL was plated on Petri dishes containing the appropriate medium with an automatic seeder (Spiral Platter Eddy Jet, I&L biosystems, Konigswinter, Germany). The plates were incubated overnight before performing a manual enumeration following the manufacturer’s instructions. 

### 2.3. Bacteria Cultivation over One Day

The growth of defined concentrations of *E. coli* (1 × 10^6^ cfu.mL^−1^) into LB at 37 °C and 190 rpm was performed for one day. Spectral measurements were performed every hour ([2,3,4,5,6] and [26–29 h]) or every 30 min ([6–13 h 30 min]).

### 2.4. CEM and Bacteria Mixture Preparation

T lymphoblasts (CEM-C1 cells, ATCC^®^ CRL-2265TM) were supplied by the French Blood Agency (EFS). They were grown in a phenol red-free RPMI-1640 medium (P04-16515, PAN-Biotech^®^, Aidenbach, Germany), supplemented with 25 mM HEPES (P05-01500, PAN Biotech^®^, Aidenbach, Germany), 10% heat inactivated FBS (10270-106, Fischer Scientific^®^, Illkirch, France) and 1% penicillin (10 kU.mL^−1^)/streptomycin (10 mg.mL^−1^) (FG101-01, TransGen Biotech^®^, Beijing, China). The cells were maintained at 37 °C in a humidified atmosphere containing 5% CO_2_.

Two mixtures of the CEM and bacteria were prepared with the following species concentrations in PBS 1X pH7.4:CEM at 1 × 10^5^ cell.mL^−1^ and *E. coli* at (1 × 10^7^, 1.7 × 10^7^, 3.8 × 10^7^, 8.7 × 10^7^ and 2 × 10^8^) bact.mL^−1^ (see Section 4.2 for the correspondence between cfu.mL^−1^ and bact.mL^−1^).CEM at 3 × 10^5^ cell.mL^−1^ and *L. lactis* at (1 × 10^7^, 2.6 × 10^7^, 6 × 10^7^, 1.4 × 10^8^ and 3.2 × 10^8^) bact.mL^−1^.

Prior to the preparation of the mixtures, the species’ concentrations were measured optically using the CEM spectral function [24] and bacteria functions (see below).

### 2.5. Spectral Absorption/Attenuation Measurements

*Preliminary remark.* Speaking strictly from an optical perspective, the spectral data presented in this paper result from two main light–matter interaction processes: absorption and diffusion (from particles of sizes similar to wavelength). Therefore, the word “attenuation“ should be preferred to “absorption” only. However, because “attenuation” is not a term commonly used in biology and/or micro-biology, the term “absorption” is used in this manuscript. Remarks concerning the choice of “absorption” data instead of transmission, transmittance or absorbance data are proposed in Section 4.1. Remarks concerning light–matter interaction processes are given in Section 4.10.

Spectral measurements of the bacteria/CEM suspensions were performed using the same experimental setup described in [23,24] (Figure 1). The spectroscopy measuring system consisted of a light source (AvaLight-DH-S-BAL, Avantes^®^, Apeldoorn, the Netherlands, supplier France) connected to a cuvette holder (Avantes, Avantes^®^, Apeldoorn, the Netherlands, supplier France, CUV-UV/VIS) via optical fibers (Thorlabs, USA, supplier Maisons Laffitte, France, M25L01). The white light source was switched on about 30 min before measurements were taken to allow for the stabilization of temperature and spectral characteristics. After propagation through the spectroscopy cuvette (Sigma, Saint Louis, MO, USA, CO793-100EA), the light was transmitted to the spectrophotometer (Ocean Optics, Orlando, FL, USA, supplier Paris, France, USB 4000 UV-VIS-ES) for spectra acquisition. Before each measurement and depending on the experiment, two types of reference spectra were acquired. For bacteria growth monitoring (Section 3.4.1), the reference spectrum was acquired using a cuvette containing LB; for all the other experiments (bacteria spectral shape description (Section 3.1), mixture analysis (Section 3.4.2) and contamination description (Section 3.4.3)), the reference spectrum was acquired with PBS. Suspensions of the bacteria alone and of the mixture, in case of contamination, were homogenized using several gentle inversions before each spectroscopy measurement. The spectra were recorded in transmission, in the wavelength range 177 nm–892 nm with a step of 0.22 nm using OceanView software (Ocean Insight, USA, supplier Paris, France).

### 2.6. Spectral Data Processing

A total of 395 bacteria spectral data were recorded into a text file and then transposed to Excel. The data obtained from transmission were converted into absorption percentages and all calculations were performed using Matlab™ R2020b software (Matlab™, Natick, MA, USA, supplier, Meudon, France). Only wavelengths between 330 nm and 860 nm were considered to remove measurements with high background noise. Artifacts due to the energetic emission peaks of the deuterium lamp were numerically removed. Regularly, the absorption spectra of neutral densities (Thorlabs, Newton, NJ, USA, NE05B and NE10B) were recorded and compared to the supplier’s data to ensure correct absorption spectra measurements. Only spectra containing useful information for the considered spectral range were kept; they approximately corresponded to spectra with absorption values between 3% and 94% at a 600 nm wavelength. Consequently, spectroscopically useful bacteria concentrations ranged from 10^6^ to 10^9^ bact.mL^−1^. Overall, 290 spectra were used to determine the absorption spectra shapes of the studied bacteria. The number of spectra for each bacterium are listed in Table 3.

In this work, the absorption spectra $Abs_{species}\left(\lambda ,C\right)$ are defined as:(1)$$Abs_{species}\left(\lambda ,C\right)=100\left(\frac{1}{T_{species}\left(\lambda ,C\right)}\right)$$
where $T_{species}\left(\lambda ,C\right)$ is the transmittance of the corresponding species.

## 3. Results

First, the method for describing the absorption spectra shapes is explained considering *E. coli* (Section 3.1). It was then efficiently applied to other ESKAPEE and *L. lactis* bacteria (Section 3.2). Therefore, for each bacteria the corresponding “bacteria function” was determined, which allowed cross-validations to be performed (Section 3.3). Results concerning *E. coli* cultivation monitoring, the simultaneous measurement of species concentrations in CEM-bacteria mixtures and the description of a potentially contaminated CEM cultivation were then obtained (Section 3.4).

### 3.1. Defining the E. coli Function

A mathematical description of the CEM absorption spectra has recently been published [24]. CEM spectra were efficiently described using two gaussian functions. Iterated fittings were used to express the evolution of each gaussian parameter with CEM concentration and define the CEM function. In the present work, iterated fittings were not applicable, and a more empirical approach was followed.

Figure 2a displays the absorption spectra of *E. coli* with concentrations ranging from 1.7 × 10^6^ to 5.4 × 10^8^ *E. coli*.mL^−1^. Spectra were efficiently fitted using the following sigmoidal function.
(2)$$Abs_{E.coli}\left(\lambda \right)=a1_{E.coli}+\frac{a2_{E.coli}}{5+exp\left\{10^{-2}\left(\lambda -a3_{E.coli}\right)\right\}}$$

Figure 2b shows the R^2^ values obtained when fitting the spectra with Equation (2). R^2^ was always greater than 0.85. However, R^2^ was even greater than 0.985 when the concentration was larger than 10^7^ *E.coli*.mL^−1^. Concentrations were expressed in terms of a decimal logarithm for mathematical reasons, explained in Section 4.3.

In Equation (2), the $a1_{E.coli}$ quantity is related to the offset of the sigmoid function, $a2_{E.coli}$ is related to its amplitude (together with the coefficient 5 in the denominator), $a3_{E.coli}$ indicates the position of the inflexion point and the coefficient 10^−2^ is related to the slope at the inflexion point. The $a1_{E.coli}$, $a2_{E.coli}$ and $a3_{E.coli}$ quantities in Equation (2) are obviously not constant values but are sub-functions evolving with the *E. coli* concentration. Defining these sub-functions is the subject of the next sections.

#### 3.1.1. Stage 1: General Spectra Shape Description Using a Sigmoidal Function

The goal of stage 1 was to determine the equations which could then be used to describe the sub-functions. Figure 3 shows the evolution of these sub-functions with *log*_10_(*C*).

Sub-functions were fitted in order to express them as functions of *E. coli* concentration. $a1_{E.coli}\left(log_{10}\left(C\right)\right)$ could be fitted with an exponential function, $a2_{E.coli}\left(log_{10}\left(C\right)\right)$ with a gaussian function and $a3_{E.coli}\left(log_{10}\left(C\right)\right)$ with an offset exponential function. Note that fitting $a3_{E.coli}\left(log_{10}\left(C\right)\right)$ is not efficient for concentrations below 10^7^ *E. coli*.mL^−1^. Equations corresponding to the *E. coli* sub-functions are given below.
(3)$$a1_{E.coli}\left(\tilde{C}\right)=p1_{a1}\cdot exp\left(p2_{a1}\left(\tilde{C}\right)\right)$$
(4)$$a2_{E.coli}\left(\tilde{C}\right)=p1_{a2}\cdot exp\left\{-{\left(\frac{\tilde{C}-p2_{a2}}{p3_{a2}}\right)}^{2}\right\}$$
(5)$$a3_{E.coli}\left(\tilde{C}\right)=p1_{a3}+p2_{a3}\cdot exp\left(p3_{a3}\left(\tilde{C}\right)\right)$$

In these equations, $\tilde{C}=log_{10}\left(C\right)$ and C is the concentration.

Finally, the complete mathematical description of the *E. coli* absorption spectrum could be written by inserting Equations (3)–(5) into Equation (2).
(6)$$Abs_{E.coli}\left(\lambda ,\;\tilde{C}\right)=p1_{a1}\cdot exp\left(p2_{a1}\left(\tilde{C}\right)\right)+\frac{p1_{a2}\cdot exp\left\{-{\left(\frac{\tilde{C}-p2_{a2}}{p3_{a2}}\right)}^{2}\right\}}{5+exp\left\{10^{-2}\left(\lambda -p1_{a3}+p2_{a3}\cdot exp\left(p3_{a3}\left(\tilde{C}\right)\right)\right)\right\}}$$

The parameters in Equation (6) are summarized in Table 4.

The values presented in Table 4 are approximative because the sub-functions (and corresponding parameters) were established independently from each other. Indeed, stage 1 made it possible to explicitly determine the concentration dependencies of the sub-functions.

#### 3.1.2. Stage 2: Determining Parameters using a Minimization Method

##### Parameters Calculation

The *E. coli’s* spectra evolution formed a concentrated surface that could then be directly adjusted with Equation (6) by simultaneously estimating the parameters’ values. A Matlab™ minimization algorithm was used to determine the set of parameters that minimized the following error function:(7)$$error=\sum_{\lambda }\sum_{\tilde{C}}{\left(Abs_{E.coli}\left(\lambda ,\tilde{C}\right)-ExpSpectra\right)}^{2}$$

Here, ExpSpectra represented the 35 absorption spectra shown in Figure 2a and $Abs_{E.coli}\left(\lambda ,\tilde{C}\right)$ is given by Equation (6).

The minimization algorithm requires a set of starting points. The approximated parameters given in Table 4 were used as the starting points for the minimization of Equation (7). Table 5 gives the parameters obtained by minimization.

##### Spectrally Measuring *E. coli* Concentrations in Stage 2

*E. coli* spectra were fitted using Equation (6) with the parameters from Table 5 to spectrally measure the *E. coli* concentrations (Figure 4).

It was observed that concentrations could be efficiently calculated only for enumeration values ($\tilde{C}$) greater than seven. This showed that the *E. coli* function could only be used within the validity range displayed in Figure 4. *E. coli* function parameters were then re-calculated considering this reduced validity range.

#### 3.1.3. Stage 3: *E. coli* Function within the Validity Range

##### Determining *E. coli* Parameters for Stage 3

The minimization method was then applied within the validity range using the starting points obtained in stage 2. The parameters obtained in stage 3 are given in Table 6. They are the final parameters for the *E. coli* function.

##### Spectrally Determining *E. coli* Concentrations in Stage 3

Figure 5 shows the results obtained from fitting the *E. coli* spectra using Equation (6) with the final parameters.

The dispersion and bias of this *E. coli* model were calculated according to the description in [24]. We recall that the “bias” is obtained by fitting the experimental data with the function Y=X+Bias. The “bias” indicates an over/under estimation of the calculated concentrations compared to the enumerated concentrations. Ideally, both should be superimposed, but dispersion (hereafter Disp.) exists (Figure 5). Disp. is calculated using a modified form of the Standard Deviation definition:(8)$$disp=\sqrt{\frac{1}{n}\sum_{i=1}^{n}{\left({\tilde{C}}_{i}^{calc}-\left({\tilde{C}}_{i}+Bias\right)\right)}^{2}}$$

Here, ${\tilde{C}}_{i}$ is the enumerated concentration of the suspension number ‘*i*’ and ${\tilde{C}}_{i}^{calc}$ is the corresponding calculated value. The dispersion and bias values of the experimental results shown in Figure 5 were given by:

Disp. = 0.14 (in log units), about 2% at center range.

Bias = −0.01 (in log units), virtually zero.

Note that the dispersion estimated when measuring the concentrations with a seeder is of the order of 0.12 in log units. Considerations regarding the dispersions and actual spectral method accuracy will be discussed in Section 4.4 and in the Appendix A. Indeed, the dispersion calculated here was not a measurement of the spectral method’s accuracy but an estimation of the spectroscopy cuvette’s variability and of the difficulty in cultivating and enumerating the bacteria.

The lower validity bound for the *E. coli* function was measured at 7.05 log units. The upper validity bound was set to the value of the maximum *E. coli* concentration used in this experiment, i.e., 8.63 log units. However, the *E. coli* function is probably valid for higher concentrations. In terms of *E. coli* concentrations, the validity range was therefore 1.12 × 10^7^ to 4.3 × 10^8^ *E.coli*.mL^−1^.

##### Graphical Representations for *E. coli*

Experimental and Theoretical Spectra were Compared

The colored surface shows the 3D representation of the *E. coli* function (Equation (6) with the final *E. coli* parameters, Figure 6). The distribution of the experimental spectra around the *E. coli* function (black curves) denotes the dispersion estimated above.

Equation (6) shows that the *E. coli* function is composed of a concentration-dependent baseline and a concentration/wavelength-dependent sigmoid, illustrated with three *E. coli* concentrations (Figure 7).

The *E. coli* function was then established empirically. It describes the *E. coli* spectra with the bacteria concentrations considered in this study. Using this function with higher concentrations may lead to unrealistic absorptions greater than 100%. The highest concentration to be used with the *E. coli* function was determined to be 4.8 × 10^8^ *E. coli*.mL^−1^, corresponding to 8.68 in log units.

This method for mathematically describing the *E. coli* absorption spectra shapes was then applied to other bacteria.

### 3.2. Generalization to Other Bacteria

#### 3.2.1. Qualitative Observations

Absorption spectra were recorded for other bacteria from the ESKAPEE group and from *L. lactis*. Examples of the spectra measured for *S. aureus*, *E. cloacae* and *L. lactis* are displayed in Figure 8.

All the spectra exhibited different but similar shapes, meaning that they could probably be described using Equation (6) with parameters corresponding to each particular bacterium. A more objective approach was taken by performing a Principal Component Analyses (PCA). The result is shown in Figure 9, where PC2 is plotted as a function of PC1. Other PCs did not contribute any further Appendix A.

The data corresponding to all the considered bacteria were situated in the same area, meaning that the spectra shapes were mathematically similar.

#### 3.2.2. Quantitative Descriptions

Since the shapes of the absorption spectra were similar, the spectral functions corresponding to each bacterium were calculated using the method presented in Section 3.1. Table 7 shows the bacteria-specific parameters calculated for each species considered in this study together with the lower and upper validity bound, the dispersions obtained with spectral measurements (Disp. M.) and with enumerations (Disp. E.).

The theoretical absorption spectra were plotted for three different bacteria concentrations (Figure 10). The spectra of *E. coli* and *L. lactis* were plotted with thicker lines for the purpose of Section 3.4.3.

It is clearly visible that, although similar, the absorption spectra strongly differed from one bacterium to another one.

### 3.3. Cross-Validation Analyses

In Section 3.2, a spectroscopy model was calculated using five cultivation sets and was tested with the same five sets. In other words, the model was evaluated using the data which was used to establish it.

A more reliable approach is to perform cross-validations. Three sets were chosen among the five available sets. These were called the “model sets” and were used to establish the spectroscopy model. The latter was applied to the two remaining “test sets” to calculate the corresponding concentrations. The process was then iterated for all possible set combinations ($C_{5}^{3}=10$, in our case). Cross-validations were performed for all the bacteria considered in this work. The results in terms of dispersions is shown in Figure 11 with box plots.

It was observed that, except for *A. baumannii*, the dispersions obtained were similar for both techniques. This aspect is discussed in Section 4.6. The dispersions calculated here were not a measurement of the spectral method accuracy but an estimation of the spectroscopy cuvettes’ variability and of the difficulty in cultivating and enumerating the bacteria.

### 3.4. Some Applications Examples

A mathematical description of the spectra shapes of bacteria absorption can be used not only to monitor bacteria cultivation in real time, but also to simultaneously measure the concentrations of species in mixtures, detect contaminants [21,22] and/or confirm a contamination occurring during cell culturing.

#### 3.4.1. Monitoring of Bacteria Cultivation without Seedings/Enumeration

The *E. coli* function was used to monitor a 30 h cultivation experiment. This cultivation was performed in a RPMI medium, which is not the normal cultivation medium for *E. coli*. Originally, the goal was to monitor a co-culture of CEM-*E. coli* since these studies were conducted in the framework of ATMP production. The choice of RPMI as the culture medium was due to the fact that priority was given to the growth of CEM.

Figure 12 shows the spectra recorded during this experiment. An additional signal around 410 nm was observed and its possible origin is discussed in Section 4.7. This additional signal was removed numerically as explained in the Appendix A. Figure 12b shows the spectra after extraction of the *E. coli* signal.

The spectra recorded during the 30 h experiment were then exploited. Results are shown in Figure 13.

A PCA analysis was performed with the spectra from Figure 12 (Figure 13a). Figure 13b shows the evolution of the *E. coli* concentration with time (red stars). The lag period, the exponential phase and the stationary phase are clearly visible. The lag time is measured in Figure 13c, where the natural logarithm of the concentration is plotted versus time and estimated at 5 h 40. This value is discussed in Section 4.8.

A black linear regression of the exponential phase, shown in Figure 13c, allowed for the determination of the bounds for the exponential fitting shown in Figure 13b (blue solid line) and for the calculation of a generation time of 1 h 40. This value will also be discussed in Section 4.8.

The accuracy of the spectral concentration measurements was estimated as it was performed in [24].
(9)$$acc=\sqrt{\frac{1}{n}\sum_{i=1}^{n}{\left(C_{calc}\left(t_{i}\right)-ExpFit\left(t_{i}\right)\right)}^{2}}\;$$

Here, $C_{calc}\left(t_{i}\right)$ refers to the *E. coli* concentration calculated at time $t_{i}$ and $ExpFit\left(t_{i}\right)$ is the value of the exponential fitting at time $t_{i}$. The estimated accuracy was:

acc = 5.8 × 10^6^ *E. coli*, which represented 1.9 % at the center concentration range.

This represented an accuracy equal to 0.0014 log units, 100 times better than the dispersion calculated in Section 3.1.3. This is because this experiment took place in a single spectroscopy cuvette, thus avoiding cuvette variability. Indeed, 1.9% should be regarded as the accuracy of the spectral measurement method.

Pure *E. coli* cultivation could then be monitored. Experiments simultaneously measuring the concentration of species in mixtures were performed.

#### 3.4.2. Simultaneous Measurement of CEM and Bacteria Mixtures

Using the additivity law of absorbances and combining the definitions of absorbance and transmittance, it can be shown that the absorption spectrum of a mixture of ‘*n*’ different species is given by Equation (9).
(10)$$Abs_{Mix}\left(\lambda ,C_{1}\ldots C_{n}\right)=100\left\{1-\prod_{i=1}^{n}\left(1-\frac{Abs_{i}\left(\lambda ,C_{i}\right)}{100}\right)\right\}$$

Two types of mixtures of mammalian cells with bacteria (CEM with *E. coli* and CEM with *L. lactis*) were prepared with the concentrations given in Section 2.4. The spectra of these mixtures are shown in Figure 14 (CEM-*E. coli* in Figure 14a, CEM-*L. lactis* in Figure 14b). PCA were performed (Figure 14c). 

As expected, both mixtures were situated between pure *E. coli* (bacteria model) and pure CEM areas. Suspensions with no bacteria were situated within the CEM area with their positions depending on the initial CEM concentrations. Because the CEM concentrations were kept constant, the trends of the mixtures were homothetic to bacteria representation.

Equation (10) was used to simultaneously calculate the CEM and bacteria concentrations (Figure 15). 

The values of the CEM concentrations were slightly dispersed as was expected for a multi-cuvettes experiment. The mean CEM concentration value was slightly lower than expected (8.7 × 10^4^ CEM.mL^−1^ instead of 1 × 10^5^ CEM.mL^−1^ in the CEM–*E. coli* mixture and 2.7 × 10^5^ CEM.mL^−1^ instead of 3 × 10^5^ CEM.mL^−1^ in the CEM–*L. lactis* mixture). This difference was due to a minor experimental imprecision and not the inefficiency of spectra fittings with Equation (10). This is discussed in Section 4.9. Bacteria concentrations were accurately calculated except for high bacteria concentrations. The reason for this has not been yet identified.

Using the additivity law of absorbances and combining the definitions of absorbance and transmittance, the concentrations of the species in the different mixtures were measured simultaneously. Equation (10) was also used to describe a potentially contaminated CEM cultivation.

#### 3.4.3. Describing a Probably Contaminated CEM Cultivation

##### Observing Contamination and Evaluating with PCA

Unusual spectra shapes were observed during a CEM culture (Figure 16a). They were similar to the shapes observed in mixtures (in the previous section) and contamination was suspected. A PCA analysis was performed to study this CEM culture (Figure 16b). The CEM concentrations in the spectroscopy cuvettes were not measured with an automated cell counter. Only the dilution factors corresponding to each cuvette were known (legend of Figure 16a).

As for the PCA performed on the mixture, the data corresponding to the unusual spectra were located between the CEM and bacteria areas. This confirmed that a contamination occurred. The next step was to measure the CEM and bacteria concentrations. The trends of mixtures were not perfectly homothetic to bacteria representation because the CEM concentration was not constant among the measurements.

##### Measuring Species Concentrations

At the time of this experiment, only *E. coli* and *L. lactis* were used in our laboratory and two hypotheses were established for the contamination’s origin. Equation (10) was used to optically measure the CEM and bacteria concentrations for both hypotheses using the CEM function and corresponding bacteria function (Figure 17).

As expected, the concentrations of the species behaved linearly. As shown in Figure 17a, the CEM concentrations were the same regardless of the contamination hypothesis. Figure 17b shows that the bacteria concentrations evolved differently depending on the hypothesis. The *E. coli* concentrations were lower than the *L. lactis* concentrations. This is because *E. coli* absorbed more than *L. lactis*, i.e., a lower concentration was required to produce the same effect.

The contaminated data shown in Figure 16b were a little bit less homothetic to bacteria representation than that in Section 3.4.2. This is because dilutions were considered in this section. Contrary to what was performed in Section 3.4.2, the CEM concentration was not constant in all the cuvettes, which explains the reduction in homotheticity.

## 4. Discussion

### 4.1. Format of the Spectroscopy Data

Optical spectra are expressed in terms of absorption measured as a percentage as already explained in [24]: “Investigations could have been conducted in any other equivalent format since absorbance (or OD), transmission, or transmittance spectra all strictly contain the same information. Our goal was to provide a method to measure concentrations in-line, without sampling, close to or inside a bioreactor. To this end, compact and low-cost components were chosen. We did not consider methods based on the use of ultra-sensitive detectors such as Photomultiplier Tubes usually used in plate readers. Therefore, data corresponding to low transmission (i.e., high OD) were not fully reliable in our case and we decided not to consider absorbance measurements. The choice between transmission and absorption was made considering that absorption spectra were easier to mathematically describe than transmission spectra.” A remark on the relevance of designating spectra “absorption” is proposed in Section 4.10.

### 4.2. Units Used for Bacteria Concentration Measurements

Bacteria concentrations are usually expressed in cfu.mL^−1^. Some bacteria tend to agglomerate and colonies may not be issued from a single organism. Moreover, enumeration only concerns living bacteria and not the total number of organisms. The actual bacteria concentration could then be underestimated. However, we assumed that 1 cfu.mL^−1^ is equivalent to 1 bact.mL^−1^. 

### 4.3. Concentrations Expressed in Decimal Logarithm

Concentrations were expressed in terms of the decimal logarithm (especially when establishing bacteria functions). The decimal logarithm was chosen for mathematical purposes. Bacteria concentrations were chosen so that the absorption spectra covered the whole range of useful values (Section 2.6, Figure 2 and Figure 8). From there, evolutions of the bacteria’s sub-functions were mathematically described from the experimental data (red stars in Figure 3).

In log units, these data were evenly distributed along the X axis. This was not the case when the concentrations were expressed in bact.mL^−1^ (Figure 18). The description of the sub-functions was facilitated when the experimental data were expressed in log units. More importantly, when expressed in bact.mL^−1^, the experimental data were mainly grouped close to the C = 0 bact.mL^−1^ vertical axis. This led to a decrease in the sub-function’s determination efficiency. Bacteria functions were then expressed in terms of the decimal logarithm.

### 4.4. Dispersion Estimations and Actual Accuracy of Spectral Measurements

The dispersions mentioned here were principally due to the variability of the spectroscopy cuvettes (already discussed in [24] and presented in the Appendix A) and to the variability of the enumeration caused by using the automatic seeder.

#### 4.4.1. Concerning Bacteria Enumeration

The dispersions measured when considering bacteria enumerations are listed in Table 7. The values were close to those spectrally measured due to the cuvettes’ variability. Seeding with an automatic seeder is highly reproducible. In this work, the enumeration was performed manually, creating a possible source of dispersion. However, we believe that the enumeration dispersion reflects how difficult it is to reproductively cultivate and subsequently enumerate bacteria. For example, model bacteria such as *E. coli* and *L. lactis* exhibited average dispersion values. High variabilities were measured with *K. pneumoniae* and *A. baumannii*. Dispersions in these cases corroborated observations made during the experiments. *K. pneumoniae* suspensions were particularly viscous while *A. baumannii* was particularly fragile, resulting in cultivation and solution homogenization difficulties.

The case of *E. faecium* should be mentioned as the dispersion was only 0.08 log units. This probably means that this bacterium can be cultivated and enumerated with a high reproducibility. For these reasons, *E. faecium* could be regarded as a “model” bacteria.

#### 4.4.2. Concerning the Accuracy of Spectral Measurements

As already explained in [24], in the case of CEM cultivation, the dispersions mentioned above should not be understood as an estimation of the spectral measurement’s accuracy. They are due to the performance of multi-cuvette experiments (see Appendix A). We recall that our main motive was to set up a sample-free, in-line and real-time measurement technique for which measurements are performed in a single cuvette. 

In this study, the accuracy of the spectral measurements can only be assessed for the experiments where only one cuvette was used. Monitoring of the *E. coli* cultivation (Section 3.4.1) showed an accuracy of 5.8 × 10^6^ *E.coli*.mL^−1^. This represents only ±1.9% at center concentration range. This value should be understood as the actual spectral measurement accuracy.

### 4.5. Validity Range of Bacteria Functions

Bacteria functions are empirical equations which can only be used within lower and upper bounds. Below the lower bound, functions are unable to properly estimate bacteria concentrations. The upper bound corresponds to the limit above which the bacteria functions produce absorptions greater than 100%. This upper bound was calculated by considering physico-mathematical considerations. It is possible that, to some extent, bacteria functions can be used with concentrations above the upper bound. In that case, the fitting would be only partial but the calculated concentrations might still be valid. However, we did not test this.

Establishing a real bacteria function, i.e., one with no upper limit, would be quite lengthy and necessitate an extensive study which is beyond the scope of this paper. Indeed, complex “light-biological entities” processes and their mutual interactions should be considered. These processes mainly involve absorption for large entities, such as cells (about 10 µm in diameter), and diffusion for smaller entities, such as bacteria (with characteristic dimensions around 2 µm).

### 4.6. Cross-Validations

Boxplots of the dispersions were obtained with cross-validations (Figure 11). Except for *A. baumannii*, the mean dispersion values were similar to those listed in Table 7. The heights of the boxplots were approximately constant for all the bacteria and the dispersion values of the enumeration remained within the span of the boxplots. This means that the dispersions were due to cuvette (constant height) and seeder variability (data within the boxplots).

Extensions of the boxplots were also constant except for *K. pneumoniae* and *A. baumannii*. This confirms the previous remarks concerning the difficulty of cultivating these bacteria and homogenizing the suspensions. Concerning *A. baumannii*, in particular, the seeder dispersion was almost three times higher than the dispersion measured optically. This emphasizes our observations relating to the fragility of this bacteria.

### 4.7. Additional Signal Observed during E. coli Cultivation Monitoring (Section 3.4.1)

The spectra recorded during the *E. coli* cultivation in RPMI showed an additional signal centered at 410 nm wavelength, which was observed only during the monitoring experiments. The origin of this signal is not yet fully understood. *E. coli* monitoring was recorded while experimenting with the CEM-*E. coli* co-cultures. Our hypothesis is that the RPMI medium could have constituted a stress factor for the bacteria, which were not grown in their ideal medium. Metabolites could have then been produced and the additional signal could be a signature of this supplementary production. Studies are now being conducted to better understand this aspect.

Note that the concentration could have been fitted considering the truncated spectra. However, the “extraction” method should be very useful once a specific study of the additional signal is performed.

### 4.8. Lag and Generation Times during E. coli Cultivation Monitoring

In Section 3.4.1, lag and generation times of 5 h 40 and 1 h 40, respectively, were measured. For the *E. coli* in optimum growth conditions, the lag time was about 53 min and the generation time was 20 min [27]. As mentioned above, the bacteria were grown in RPMI instead of a LB medium for monitoring. The use of this medium not only induced an additional signal at 410 nm wavelength but also increased the lag and generation times because of the stress induced on the bacteria [28]. Similar lag and generation times were observed in other experiments. Moreover, spectral concentration measurements proved to be a convenient tool for studying such bacteria characteristics because of their ease of use.

### 4.9. Robustness of Simultaneous Specie Concentration Measurements

The simultaneous concentration measurement of two species was demonstrated in Section 3.4.2 and Section 3.4.3 using Equation (10).

Concerning the results discussed in Section 3.4.2, the pure CEM concentrations measured spectrally were slightly lower than expected. As already mentioned, this was due to a minor experimental imprecision and not to an inefficiency of spectra fittings with Equation (10). Indeed, it could be argued that measuring a pure CEM suspension with the “mixture” Equation (10) would lead to wrong values for the CEM concentrations because of the influence of the bacteria’s contribution in the fitting function. This was verified using tube 1 of the CEM–*L. lactis* mixture, for which the theoretical CEM concentration was 3 × 10^5^ CEM.mL^−1^. A calculation for tube 1 of the CEM–*E.coli* mixture was not considered as the CEM concentration was lower than the detection limit of the CEM equation [24].

In order to estimate the robustness of the CEM concentration measurement in a mixture, the latter was calculated considering all the bacteria functions established in this paper. We obtained the following average value for CEM concentration:CEM_Mix_ = 2.6 × 10^5^ ± 4.8 × 10^3^ CEM.mL^−1^, i.e., ±1.9%

This shows that the robustness of the mixture function was not perturbed by the bacteria contributions, regardless of the species. However, calculations with the pure CEM function led to a value of 2.7 × 10^5^ CEM.mL^−1^ (Section 3.4.2). Indeed, use of the mixture function instead of the pure CEM function when measuring pure CEM suspensions slightly underestimated the actual CEM concentration. However, the underestimation was only about 3.7%, which was more than acceptable in this context (see discussion in [24]).

It is also remarkable that, regardless of the bacteria considered, the CEM concentrations were measured correctly (Section 3.4.3). This was because the shapes of the bacteria’s and CEM’s spectra were relatively different, as shown in the PCA analysis. The simultaneous monitoring of co-cultures is likely to be robust only if the shapes of the spectra are sufficiently different, allowing for the fitting using Equation (10) to work correctly.

### 4.10. Light–Matter Interaction Processes

The main light–matter interaction processes involved in this study are absorption and scattering. Thousands of molecules, proteins or other entities contained in suspensions are susceptible to light absorption. Among them, cytochromes have been studied for decades [29,30]. They principally absorb light between 500 and 600 nm wavelength. Mainly present in mammalian cells, they contributed to the shape of the absorption spectra we measured for the T-cells [24]. They are almost absent in bacteria [31]. However, absorption is not the main light–matter interaction process occurring in bacteria suspensions.

Indeed, scattering theory explains that for large particles (such as T-cells, about 10 µm), scattering occurs in the Mie regime, in which light propagates forward almost independently of the wavelength. Light attenuation is mostly due to absorption as shown in [24]. Bacteria are roughly the same size as the visible wavelengths. In this case, the scattering efficiency evolves as a function of 1/λ. This is the predominant light–matter interaction process that occurred in our bacteria suspensions and explains the shape of the absorption spectra we measured. Indeed, the low attenuation measured at long wavelengths is due to the fact that light coupling between the optical fibers of the experimental set-up is stronger for less scattered wavelengths than for highly scattered (short) wavelengths.

Suspensions do not only contain mono-dispersed particles, but a collection of varying-sized entities. Indeed, the metabolism of bacteria produces sub-wavelength particles (vesicles, for example). To exactly describe the light–matter interaction processes would require considering pure absorption, the Mie 1/λ regime (bacteria) and also the Rayleigh 1/λ^4^ regime (sub-wavelength entities), which would be quite complicated and is beyond the scope of this paper. This is why an empirical description of what we call “absorption spectra” is proposed in this paper and is convenient for our purpose. Note that “absorption” is somewhat a misuse of language and a more proper term would be “attenuation,” as it conveys both absorption and scattering effects.

### 4.11. Position of Our Studies and Model to Others

The method presented here could be easily adapted to a sample-free, seeder-free and real-time bacteria monitoring system such as that already proposed in [22]. Measurements were based on spectroscopy performed on a large volume which led to a high concentration-determination accuracy. The large volume required in this investigation could have been considered a drawback of this method, but it is no longer an issue with the use of a derivation loop for real-time monitoring.

The derivation of bioreactor content may possibly be envisaged with methods employing microfluidic chips. Most of the time these methods, such as SPR or QCM, require immobilizing bacteria on a functionalized surface [17]. This could constitute a drawback as the ligands used for immobilization may be reinjected into the bioreactor after measurement. In addition, such sensors may rapidly saturate, which would require breaking the close-loop environment for sensor cleaning, regeneration or replacement.

Label-free methods such as cytometry [32], lens-free imaging [33] or impedance-based methods [4] may be an alternative to sensor saturation. However, they seem difficult to implement in a real-time measurement device. Obviously, methods based on suspension sampling and/or subsequent bacteria cultivation cannot be considered as candidates for a real-time monitoring device [1,34]. Commercially, the Bactobox device [19] is probably the closest to a real-time measurement but still requires the sampling of a small suspension volume.

Bacteria detection aims to access very low micro-organism concentrations, which is the case when very small sample volumes are considered. For example, the method based on IC 3D allows the detection of 1 bact.mL^−1^ [15]. Bacteria detection is often associated with bacteria strain identification, which can be performed using Raman spectroscopy, for example [35], but this is not within the scope of this paper. To summarize, very low limits of detection are required when the goal is to detect bacteria in cell cultures or in the agri-food industry. The mathematical methods described in this paper have not been established to detect a single bacterium in 1 mL suspensions. The goal is to propose a tool able to monitor bacteria concentration during cultivation in real-time and without sampling. 

Our mathematical methods, although a bit complex, allow real-time and sampling-free operations and the simultaneous measurement of two species (at least cells and bacteria), irrespective of their concentrations within a mixture. Very few previous studies have mentioned simultaneous detection methods which could potentially be transformed into a real-time measurement technique. In one study combining Raman spectroscopy and advanced signal processing, the detection of the effects of contamination on cell cultures [36] was performed without an identification of the contaminant. In the present study, we demonstrated the simultaneous measurement of bacteria and mammalian cell concentration; this is only possible in cases with relatively different absorption spectra shapes for the fitting to be efficient. This mathematical method could be employed for real-time contamination detection as we conceptually proposed in [21,22].

## 5. Conclusions

This paper highlighted a mathematical description of the shape of bacteria absorption spectra which was subsequently used to measure bacteria concentrations. This had the supplementary advantage of also working in the case of a co-culture.

Our model allowed the monitoring of an *E. coli* cultivation over 30 h in a single spectroscopy cuvette with an accuracy below 2%. Lag and generation times were easily calculated, which suggests that spectroscopy measurements may be an easy method for studying these aspects. A mixture function allowed for the simultaneous measurement of the concentration of both species in a mammalian and bacteria mixture, and was also efficiently used for describing a probably-contaminated CEM culture. 

In addition to having a high accuracy and a potential application to co-culture monitoring, the use of white light spectroscopy can easily be integrated into a sampling-less, seeder-less, closed-environment and real-time measurement device, which is crucial for biosimilar production. In this case, the mathematical descriptions of mixtures may be exploited to detect possible contamination during mammalian cell cultivation.

## Figures and Tables

**Figure 1 sensors-23-04325-f001:**
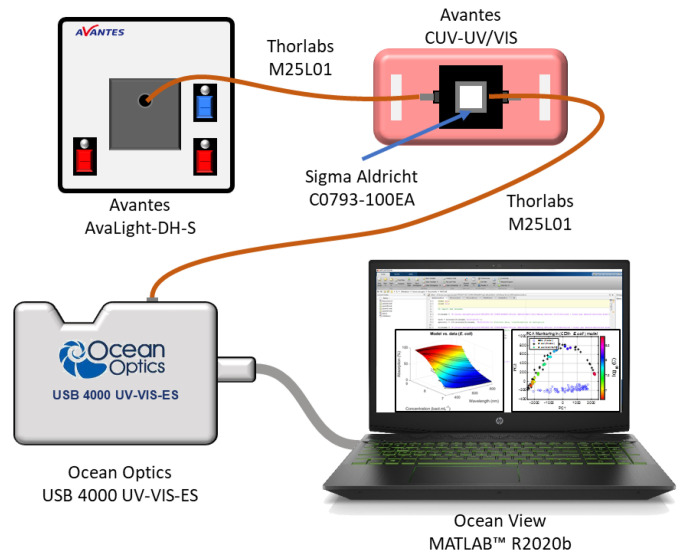
Experimental setup used for measuring absorption spectra (adapted from [23,24]).

**Figure 2 sensors-23-04325-f002:**
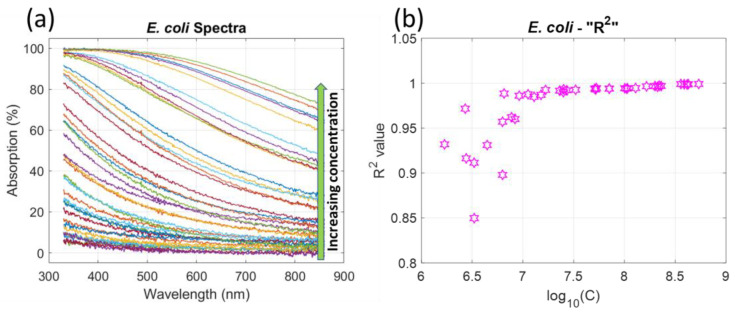
*E. coli* spectral data. (**a**) Absorption spectra for concentrations ranging from 1.7 × 10^6^ to 5.4 × 10^8^ *E. coli*.mL^−1^. (**b**) R^2^ values obtained when fitting spectra with Equation (2).

**Figure 3 sensors-23-04325-f003:**
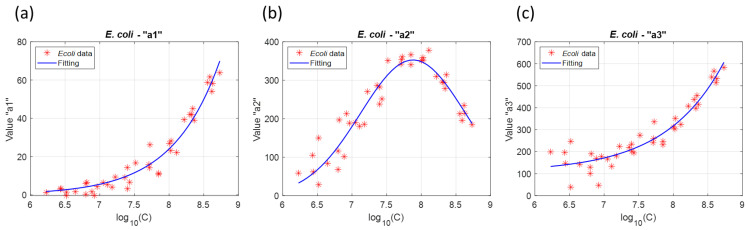
Evolution of *E. coli* sub-functions: (**a**) sub-function $a1_{E.coli}\left(log_{10}\left(C\right)\right)$, (**b**) sub-function $a2_{E.coli}\left(log_{10}\left(C\right)\right)$ and (**c**) sub-function $a3_{E.coli}\left(log_{10}\left(C\right)\right)$. The mathematical description of the evolution of the sub-function with *log*_10_(*C*) are plotted in blue solid lines.

**Figure 4 sensors-23-04325-f004:**
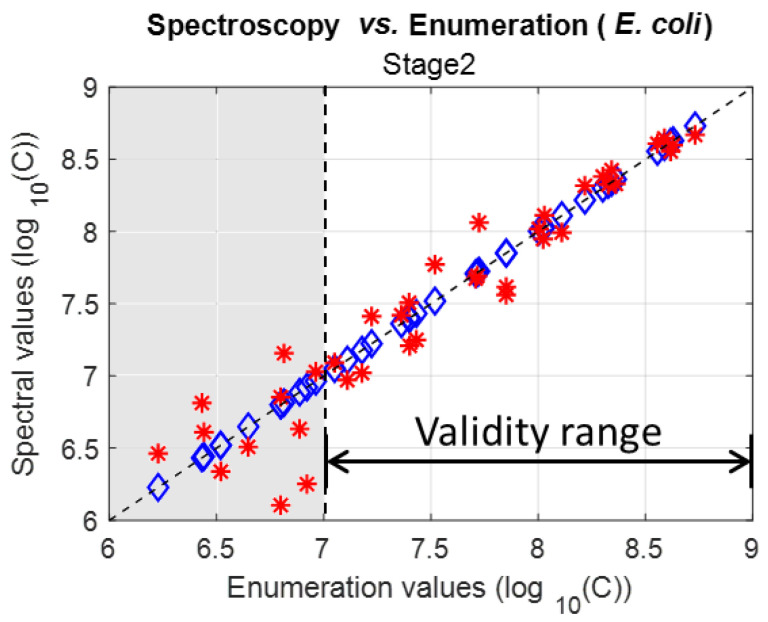
Spectrally measured concentration vs. measured concentrations using parameters at stage 2 (Blue diamond: measured concentrations (situated on the Y = X line), red stars: calculated concentrations).

**Figure 5 sensors-23-04325-f005:**
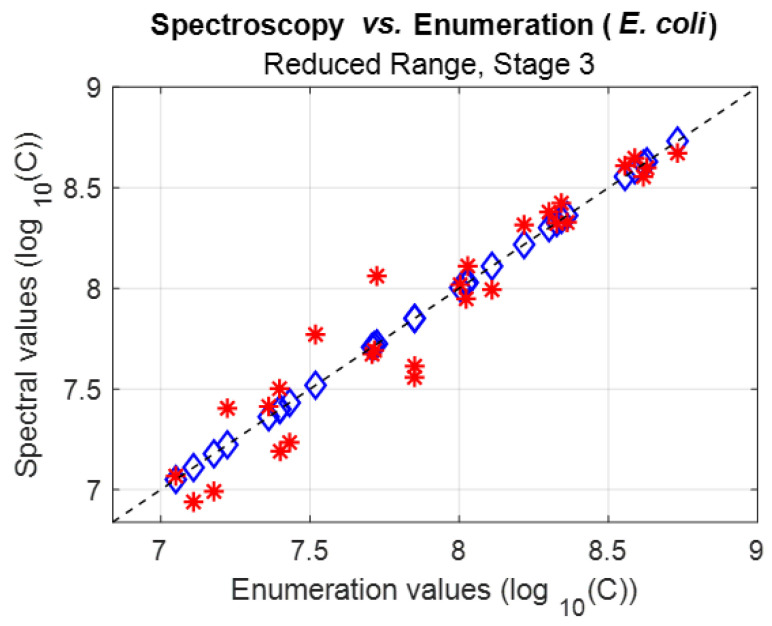
Spectrally measured concentration vs. measured concentration using the final parameters within the validity range (Blue diamond: measured concentrations (situated on the Y = X line), red stars: calculated concentrations).

**Figure 6 sensors-23-04325-f006:**
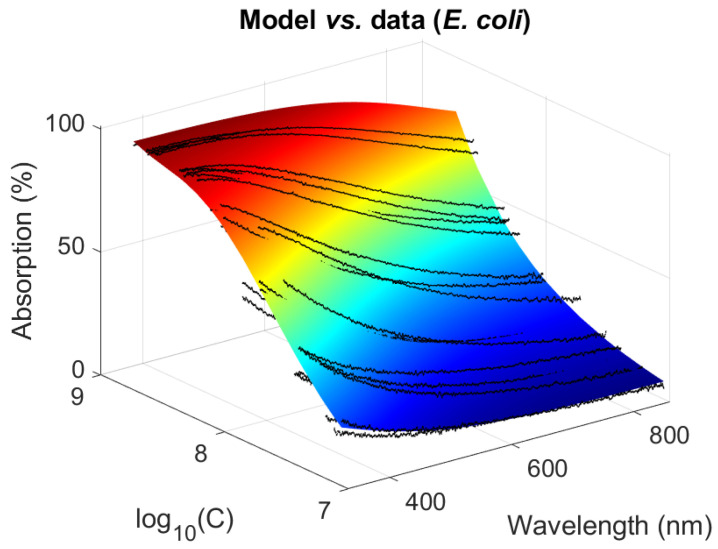
A 3D representation of *E. coli* experimental data and function (Black lines: experimental spectra, colored surface: *E. coli* function).

**Figure 7 sensors-23-04325-f007:**
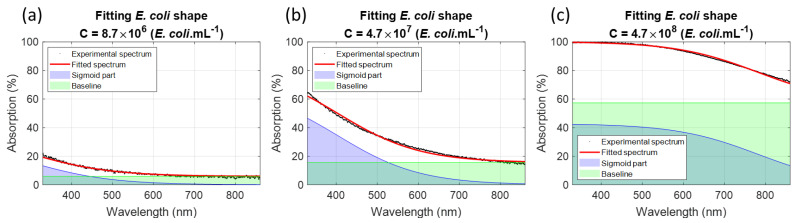
Examples of *E. coli* spectra fittings (Black dots: experimental data, red lines: fitted spectra, green surfaces: concentration-dependent baseline and blue surfaces: concentration/wavelength-dependent sigmoid). Examples given for: (**a**) 8.7 × 10^6^ *E.coli*.mL^−1^, (**b**) 4.7 × 10^7^ *E.coli*.mL^−1^ and (**c**) 4.7 × 10^8^ *E.coli*.mL^−1^.

**Figure 8 sensors-23-04325-f008:**
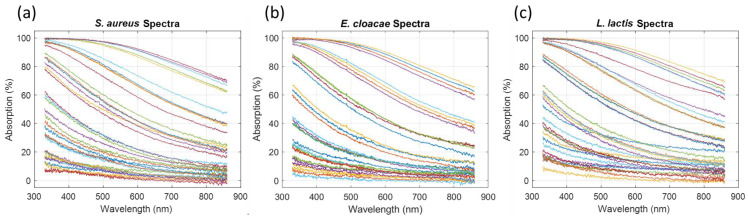
Examples of spectra recorded for other bacteria: (**a**) *S. aureus*, (**b**) *E. cloacae* and (**c**) *L. lactis*.

**Figure 9 sensors-23-04325-f009:**
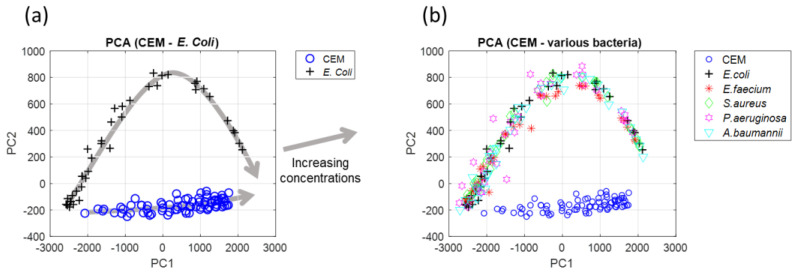
Principle Component Analyses. (**a**) PCA with CEM and *E. coli* only, showing the direction of increasing concentration (grey arrows). (**b**) PCA with CEM, *E. coli* and various bacteria.

**Figure 10 sensors-23-04325-f010:**
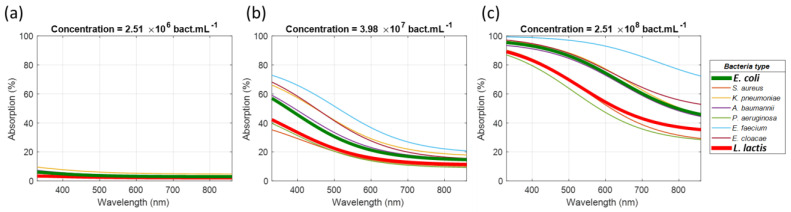
Theoretical spectra calculated for each bacterium and for different concentrations: (**a**) 2.51 × 10^6^ bact.mL^−1^, (**b**) 3.98 × 10^7^ bact.mL^−1^ and (**c**) 2.51 × 10^8^ bact.mL^−1^.

**Figure 11 sensors-23-04325-f011:**
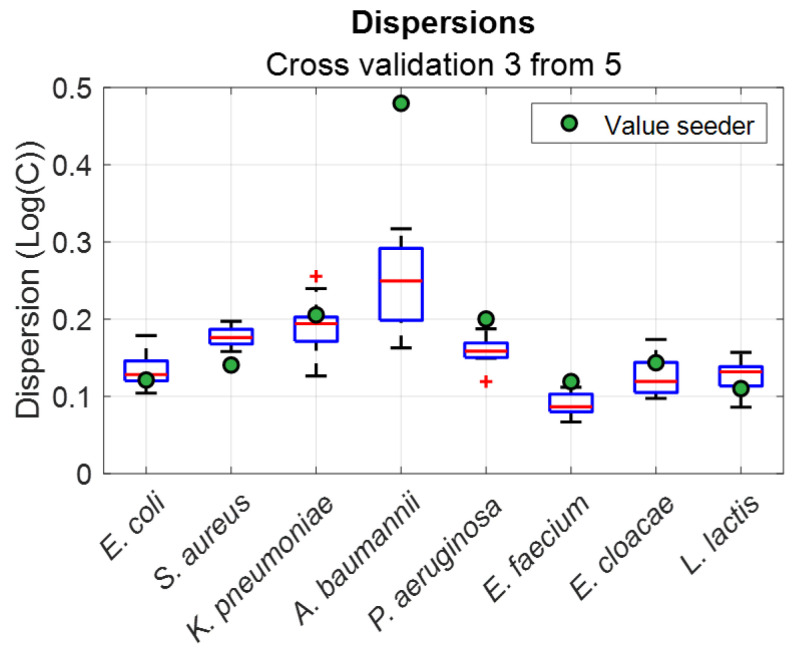
Box plots representing the dispersions calculated using cross-validations. Green circles represent the dispersion obtained with the seeder.

**Figure 12 sensors-23-04325-f012:**
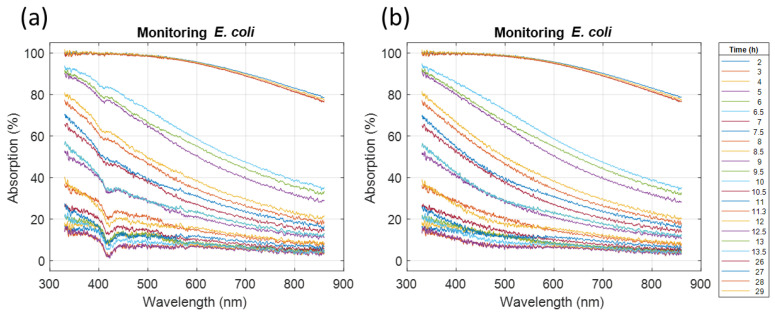
*E. coli* absorption spectra recorded during a 30 h cultivation experiment. (**a**) Raw spectra containing information from bacteria and an additional signal at 410 nm. (**b**) Extraction of the *E. coli* spectra.

**Figure 13 sensors-23-04325-f013:**
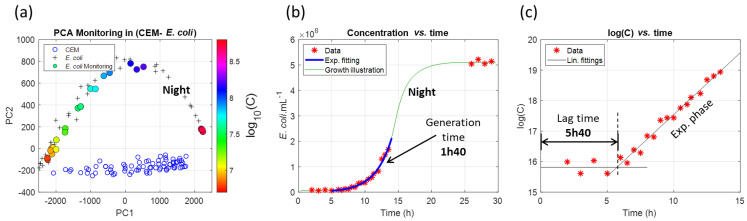
Cultivation of *E. coli* for 30 h. (**a**) PCA with color-coded spectrally measured concentrations (color bar expressed in decimal log units). (**b**) Spectrally measured concentrations past the 30 h period and estimation of the generation time (red stars: experimental data, blue line: fitting during the exponential phase, green line: growth illustration using Spline fitting). (**c**) Lag time measurement (red stars: experimental data, black line: linear regressions).

**Figure 14 sensors-23-04325-f014:**
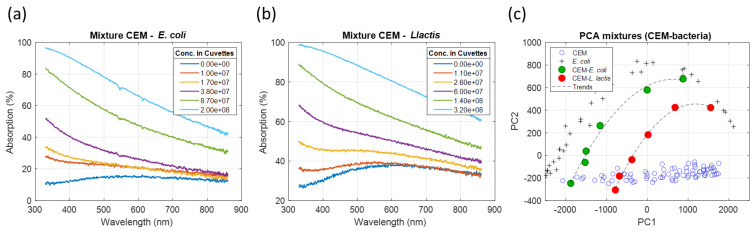
Analysis of CEM–bacteria mixtures. (**a**) CEM–*E. coli* mixture’s spectra. (**b**) CEM–*L. lactis* mixture’s spectra. (**c**) PCA of both mixture sets.

**Figure 15 sensors-23-04325-f015:**
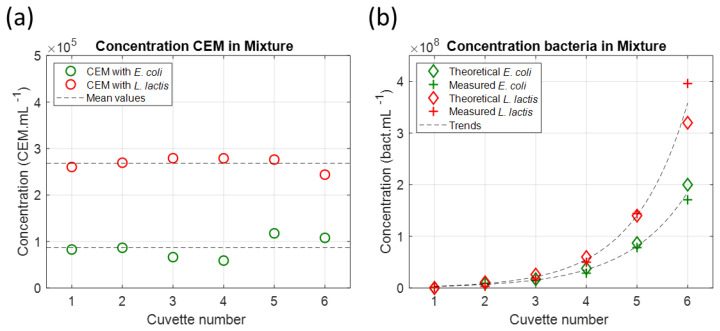
Simultaneous spectral measurement of species concentrations in mixtures. (**a**) CEM concentrations. (**b**) Bacteria concentrations (Green data: *E. coli*, red data: *L. lactis*).

**Figure 16 sensors-23-04325-f016:**
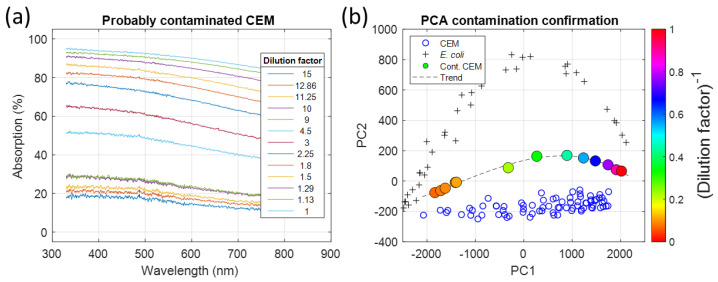
Suspicion of contamination. (**a**) Unusual spectra shapes. (**b**) PCA on these spectra (Color coding: inverse of the dilution factor. The color coding corresponds to the inverse of the dilution factor for convenience).

**Figure 17 sensors-23-04325-f017:**
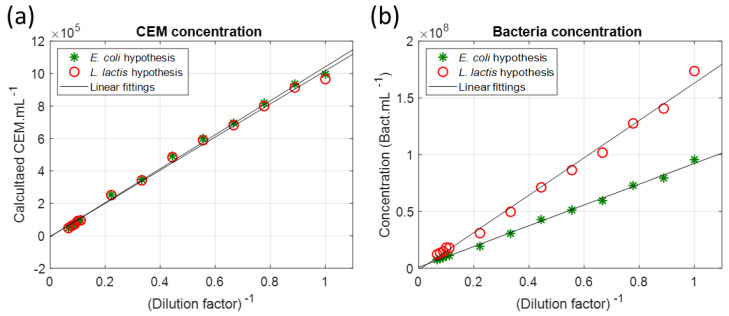
Simultaneous measurement of species concentrations. (**a**) CEM concentration as a function of the dilution factor. (**b**) Bacteria concentrations (Red stars: results for the *E. coli* hypothesis, blue circles: results for the *L. lactis* hypothesis).

**Figure 18 sensors-23-04325-f018:**
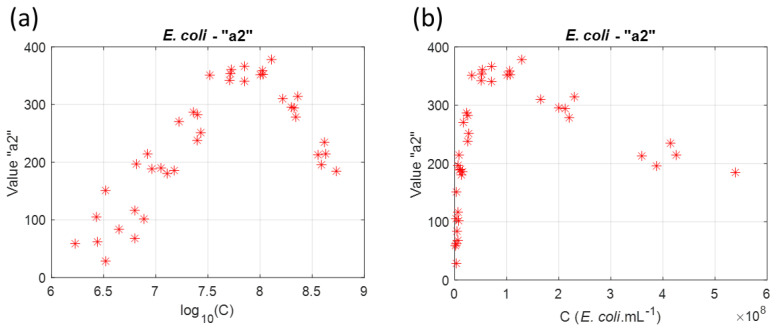
Expressing concentration in log units or in bacteria.mL^−1^. (**a**) Abscissa in log units. (**b**) Abscissa in bacteria.mL^−1^.

**Table 1 sensors-23-04325-t001:** Bacteria strains, suppliers and optimal growth temperature.

Bacteria Name	Strain	Company/Reference	Optimal Growth T °C
*Escherichia coli DH5α*	18265017	Fischer Scientific™, Illkirch, France	37 °C
*Klebsiella pneumoniae*	ATCC^®^ 13883	LGC Standard Ltd., Teddington, UK	37 °C
*Enterococcus faecium*	ATCC^®^ 8459	LGC Standard Ltd., Teddington, UK	26 °C
*Acinetobacter baumannii*	ATCC^®^ 19606	DSMZ, Braunschweig, Germany	37 °C
*Staphylococcus aureus*	ATCC^®^ 35556	DSMZ, Braunschweig, Germany	37 °C
*Pseudomonas aeruginosa*	PAO1	[26]	37 °C
*Enterobacter cloacae*	ATCC^®^ 13047	Oxoid, Fisher Scientific™	30 °C
*Lactococcus lactis*	NZ9000	NIZO	32 °C

**Table 2 sensors-23-04325-t002:** Culture media and buffer suppliers.

Name	Company	Reference
LB Broth	Difco™ (Fisher Scientific™)	241420
M17 Broth (M17B)	Oxoid (Fisher Scientific)	CM0817
M17 Agar (M17A)	DSMZ, Germany	CM0785
TSA	Oxoid (Fisher Scientific)	PO5012A
TSB	See [26]	CM1065T
Petri dishes	Starstedt, Numbrecht, Germany	82.1184.500
NaCl 0.85%	Dutscher, Bernolsheim, France	994004

**Table 3 sensors-23-04325-t003:** Number of exploited spectra per species.

Bacteria	*Escherichia* *coli*	*Staphylococcus* *aureus*	*Klebsiella* *pneumoniae*	*Acinetobacter* *baumannii*	*Pseudomonas* *aeruginosa*	*Enterococcus* *faecium*	*Enterobacter* *cloacae*	*Lactococcus* *lactis*	Total
**Enumeration (n=)**	55	46	50	45	53	40	42	64	**395**
**Exploited spectra (n=)**	40	35	35	36	35	35	39	35	**290**

**Table 4 sensors-23-04325-t004:** List of *E. coli* parameters obtained at the end of stage 1.

Parameters *E. coli* Stage 1	p1_a1_	p2_a1_	p1_a2_	p2_a2_	p3_a2_	p1_a3_	p2_a3_	p3_a3_
**Value**	1.99 × 10^−4^	1.46	353	7.88	1.07	107	1.65 × 10^−2^	1.18

**Table 5 sensors-23-04325-t005:** List of *E. coli* parameters obtained at the end of stage 2.

Parameters *E. coli* Stage 2	p1_a1_	p2_a1_	p1_a2_	p2_a2_	p3_a2_	p1_a3_	p2_a3_	p3_a3_
**Value**	3.92 × 10^−4^	1.37	359	7.93	1.02	73.7	3.49 × 10^−2^	1.11

**Table 6 sensors-23-04325-t006:** Final list of *E. coli* parameters.

Final *E. coli* Parameters	p1_a1_	p2_a1_	p1_a2_	p2_a2_	p3_a2_	p1_a3_	p2_a3_	p3_a3_
**Value**	6.47 × 10^−4^	1.31	360	7.9	1.08	36.9	7.21 × 10^−2^	1.04

**Table 7 sensors-23-04325-t007:** List of parameters, bounds and dispersions for the bacteria considered in this study.

*Bacteria*	*Escherichia* *coli*	*Staphylococcus* *Aureus*	*Klebsiella* *pneumoniae*	*Acinetobacter* *baumannii*	*Pseudomonas* *aeruginosa*	*Enterococcus* *faecium*	*Enterobacter* *cloacae*	*Lactococcus* *lactis*
p1_a1_	6.47 × 10^−4^	7.45 × 10^−4^	5.67 × 10^−3^	1.28 × 10^−3^	2.89 × 10^−4^	1.86 × 10^−4^	1.32 × 10^−4^	3.25 × 10^−4^
p2_a1_	1.31	1.25	1.05	1.23	1.36	1.52	1.53	1.37
p1_a2_	360	357	346	343	361	330	378	343
p2_a2_	7.9	8.35	7.96	7.9	8.23	7.77	7.86	8.13
p3_a2_	1.08	0.92	1.14	1.15	1.02	0.81	0.87	0.97
p1_a3_	36.9	149	17.6	−9	76.4	261	232.9	135.3
p2_a3_	7.21 × 10^−2^	3.45 × 10^−2^	1.7	6.92 × 10^−1^	3.1 × 10^−1^	2.51 × 10^−4^	3.18 × 10^−4^	2.32 × 10^−3^
p3_a3_	1.04	1.05	0.67	0.78	0.81	1.67	1.6	1.37
Lower	7.05	7.32	7.06	6.73	7.18	6.54	6.66	7.01
Upper	8.68	9.1	8.87	8.78	9.02	8.43	8.54	8.83
Disp. M.	0.14	0.18	0.22	0.35	0.16	0.08	0.13	0.13
Disp. E.	0.2	0.14	0.21	0.48	0.2	0.12	0.14	0.11

## Data Availability

Research data are available by request to the corresponding author.

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
