# Peer review of "Absorption/Attenuation Spectral Description of ESKAPEE Bacteria: Application to Seeder-Free Culture Monitoring, Mammalian T-Cell and Bacteria Mixture Analysis and Contamination Description"

_sensors, 2023, doi:10.3390/s23094325_

Round 1
Reviewer 1 Report
This is an important study on bacteria/mammalian cells concentration monitoring. The manuscript is well written. The paper may be accepted for publication after a revision that would address these comments:
1. Although the author state this is a absorption technique, there is no information in the manuscript on chromophores in these bacteria and mammalian cells. Please, provide a list of major chromophores and their absorption spectra in this spectral range (330 – 860 nm). A discussion of the major chromophores contribution to the absorption spectra of these bacteria and cells would be very helpful.
2. Bacteria and mammalian cells are strong light scatterers. Scattering can influence the spectral transmission measurements performed in this study. Typically, scattering can have a higher contribution to light attenuation in such suspensions compared to absorption. However, there is no analysis and discussion of scattering in this manuscript. Please, provide this analysis to show influence of scattering on these measurements.
3. Decrease of “Absorption” in Fig. 2a with wavelength may be associated with lower scattering at longer wavelengths (that is typical for dependance of scattering on wavelength in this spectral range). Hence “Attenuation” is more correct than “Absorption” because attenuation is a combination of absorption and scattering that directly relates to transmission through these suspensions. The experimental setup used in this study (Fig. 1) measured transmission/attenuation (not absorption per se). Actually, the “absorption” spectra in Fig. 2a could be scattering spectra of these suspensions because scattering is stronger than absorption for cells in this spectral range.
Author Response
Response to reviewer #1, manuscript 2263849
The authors would like to thank the reviewer for the attention paid to our manuscript and for improvement remarks, now included in the text.
Preliminary remark.
Before to answer to reviewer’s comments and actions made to take remarks into account, we would like to recall that the goal of this paper is to provide microbiologists with optical based methods to monitor bacteria growth and to simultaneously monitor concentrations of biological cells in mixtures (description of contaminated cell culture for example). Equations established to this end are empirical and no attempts were made to establish them on a purely bio-photonic base. However, addressing reviewer’s comments greatly improves our paper. Below, we explain how this improvement is considered.
Reviewer’s comments and tentative answers.
Reviewer comments address two main aspects. There will be one answer for remark 1 and one answer for remarks 2 and 3.
- “Although the author state this is an absorption technique, there is no information in the manuscript on chromophores in these bacteria and mammalian cells. Please, provide a list of major chromophores and their absorption spectra in this spectral range (330 – 860 nm). A discussion of the major chromophores contribution to the absorption spectra of these bacteria and cells would be very helpful.”
This remark concerns which molecules or macro-molecules present in bacteria are susceptible to absorb light. It is asked to list these entities and to propose a discussion on this aspect. In fact, there are thousands of such entities in biological cells. Establishing a list of them is almost impossible and determining their individual contribution to the absorption spectra we measure would be extremely difficult and somehow beyond the scope of this paper. However, adding a discussion on this aspect improves the paper in the sense that, together with remarks about the light-matter interaction processes occurring in bacteria solution, it allows to explain the shape of the absorption spectra we measured with T-cells (reference [24]) and why shapes of bacteria are different.
Action made: we mention that cytochromes have been studied for decades and are mainly absorbing between 500 and 600 nm wavelength. Cytochromes are mainly present in mammalian cells and contribute to the shape of the T-cells absorption spectra we presented in a previous paper (ref [24]). Shape of the absorption spectra of bacteria are mostly due to a light-matter interaction process different from pure absorption. This is part of the new section 4.10 (revised section numbering) we added to this end and titled “Light-matter interaction processes”.
- “Bacteria and mammalian cells are strong light scatterers. Scattering can influence the spectral transmission measurements performed in this study. Typically, scattering can have a higher contribution to light attenuation in such suspensions compared to absorption. However, there is no analysis and discussion of scattering in this manuscript. Please, provide this analysis to show influence of scattering on these measurements.”
- “Decrease of “Absorption” in Fig. 2a with wavelength may be associated with lower scattering at longer wavelengths (that is typical for dependance of scattering on wavelength in this spectral range). Hence “Attenuation” is more correct than “Absorption” because attenuation is a combination of absorption and scattering that directly relates to transmission through these suspensions. The experimental setup used in this study (Fig. 1) measured transmission/attenuation (not absorption per se). Actually, the “absorption” spectra in Fig. 2a could be scattering spectra of these suspensions because scattering is stronger than absorption for cells in this spectral range.”
Much more informative are these two remarks concerning the light-matter interaction processes which exists in biological solutions. Among them, the most important ones, for what we are concerned, are absorption and scattering. Theory explains that for large particles (like T-cells, about 10 µm), scattering occurs in the Mie regime for which light mostly propagates forward almost independently of the wavelength. Light attenuation is mostly due to absorption. This was the case in paper [24] which describes the shape of T-cells absorption spectra and showed absorption maxima between 500 and 600 nm (in accordance with the absorption bands of cytochromes).
In the present paper, bacteria are roughly of the same size as the visible wavelengths. In this case, the scattering efficiency evolves as a function of 1/l. As reviewer mentions, this is the predominant light-matter interaction process occurring in our bacteria suspensions and explains the shape of the absorption spectra we measured. Indeed, the low attenuation measured at large wavelengths is due to the fact that light coupling between optical fibers set-up is stronger for less scattered wavelengths than for highly scattered (small) wavelengths. The reviewer is also right when explaining that spectra should be called scattering spectra instead of absorption spectra. It is also true that, in this case, the term “attenuation” should be preferred to “absorption”. However, we prefer continuing talking about absorption since this term is usually employed by readers this paper is addressed to.
Action made: we explain these interesting aspects proposed by the reviewer in the new section 4.10. We also explain that suspensions do not only contain mono-dispersed particles but a collection of size varying entities. Indeed, bacteria metabolism produces sub-wavelength particles (vesicles for example). Therefore, exactly describing light-matter interaction processes would imply considering pure absorption, Mie 1/l regime (bacteria) and also Rayleigh 1/l4 regime (sub-wavelength entities) which would be quite complicated and beyond the scope of this paper. This is why an empirical description of what we call “absorption spectra” is proposed in this paper and convenient for our purpose (already shortly mentioned in section 4.5). We also mention that “absorption” is somehow a misuse of language because a more proper term would be “attenuation” as it groups absorption and scattering effects. One sentence has been added to section 4.1 to draw readers attention to this aspect.

Reviewer 2 Report
The reviewed article is devoted to mathematical description and analysis of the absorption spectra shapes of the ESKAPEE group bacteria, to enumerate and monitor bacteria cultivation. The authors claim simultaneous concentration measurements of both species in mammalian and bacteria mixture with an accuracy below 2%. The aim of the work is clearly formulated and the description of results as well as conclusions are very well structured and focussed. Although the topic is quite important, there are many areas of weakness where the authors should pay attention. I suggest that the authors rewrite the work by making the below corrections, shortening it, and then resubmitting it to MDPI.
Detailed remarks:
a. No comments on the abstract
b. Taking into account that the submission was made to MDPI the authors follow the MDPI guidelines, to cite references mostly recent publications.
Lower case or upper case?
Line 88: Gram- or Gram+ bacteria
Line 127: gram+ and gram- bacteria
c. Material and Methods, 2.2-Bacteria enumeration, line 162.
I suggest replacing the ‘reason 2’ with the actual reason, “following a serial dilution of reason 2 in PBS 1X”.
d. The Spectral absorbtion measurements section needs to be re-written, to specify the reference spectra (not only “a reference spectrum”). This paragraph doesn’t make sense: “Before each measurement, a reference spectrum was acquired using a cuvette containing: LB for bacteria growth monitoring (section 3.4.1) and PBS for bacteria spectral shape description (section 3.1), mixture analysis (section 3.4.2) and contamination description (section 3.4.3).”
The cuvette doesn’t contain mixture analysis and contamination description.
e. Eqn 1 (line 229) doesn’t match any of the equations in the Graphical Abstract picture, row 1 or 2.
f. Lines 231-234, why do the authors use the PCA and what is the meaning of "running a second PCA"? Authors should introduce the meaning of PCA1 and PCA2.
g. Again, what's the relation between eqn. 1, eqn. 2, and the eqns (first and second rows) in the Graphical Abstract?
h. I suggest that all parameters piai to have the ‘ai’ written as subscript.
i. I suggest sections 3.1.1 and 3.1.2 should go in a supplementary document.
j. The Extraction of the E.coli spectra section with figure 13 could go in the supplementary document.
k. Sections 4.1 to 4.3 could go in the supplementary document as well.
The paper can in principle be accepted after revision based on the above comments.
Author Response
Response to reviewer #2, manuscript 2263849
The authors would like to thank the reviewer for the attention paid to our manuscript and for improvement remarks, now included in the text.
- “Taking into account that the submission was made to MDPI the authors follow the MDPI guidelines, to cite references mostly recent publications.
Lower case or upper case?
Line 88: Gram- or Gram+ bacteria
Line 127: gram+ and gram- bacteria”
The right spelling is Gram+ or Gram-.
Action made: line 127 has been corrected.
- “Material and Methods, 2.2-Bacteria enumeration, line 162.
I suggest replacing the ‘reason 2’ with the actual reason, “following a serial dilution of reason 2 in PBS 1X”.”
We agree.
Action made: modification has been made.
- “The Spectral absorbtion measurements section needs to be re-written, to specify the reference spectra (not only “a reference spectrum”). This paragraph doesn’t make sense: “Before each measurement, a reference spectrum was acquired using a cuvette containing: LB for bacteria growth monitoring (section 3.4.1) and PBS for bacteria spectral shape description (section 3.1), mixture analysis (section 3.4.2) and contamination description (section 3.4.3).”
The cuvette doesn’t contain mixture analysis and contamination description.”
We acknowledge that we have been unclear.
Action made: text has been modified.
- “Eqn 1 (line 229) doesn’t match any of the equations in the Graphical Abstract picture, row 1 or 2.”
Equation 1 shows the relationship between the transmittance and the absorption expressed in %. This is why it is normal that it does not have to match any of the equation written in the graphical abstract.
Action made: none.
- “Lines 231-234, why do the authors use the PCA and what is the meaning of "running a second PCA"? Authors should introduce the meaning of PCA1 and PCA2.”
During our work, we started running a PCA with T-cells and E. coli. We used to call it “model”. Only after we considered other bacteria and ran PCA with the “model” spectra + 1 spectrum of another bacterium. Doing this, data representing CEM and E. coli remain “roughly” situated at the same coordinates in the PC1-PC2 space. Since we did not use PCA for calculation, this was enough for visual representations. We acknowledge we have not been clear. Indeed, since this explanation would lengthen the paper without adding important information, it is probably better not to mention this aspect.
Action made: mention to extra PCA has been removed (lines 231-234 initial version). Figure 9 and caption have been updated. Figure 13 (revised version) has been updated. Text in lines 441 to 444 and line 776 (initial version) has been modified.
- “Again, what's the relation between eqn. 1, eqn. 2, and the eqns (first and second rows) in the Graphical Abstract?”
Equation (1) already addressed (point 4). Equation (2) and equation in second row of the graphical abstract mean the same. We have little room in the graphical abstract and writing equation in the second row the same way as equation (2) in the text would have taken too much space. Both equations are the same if we understand n=2, G1= and G2=. The same holds for equations in the second row of the graphical abstract, F1 and F2 referring to their literal expressions in [24].
Action made: none.
- “I suggest that all parameters piai to have the ‘ai’ written as subscript.”
We agree.
Action made: modification have been done accordingly.
- Reviewer’s suggestions to shorten the paper.
8.1. “I suggest sections 3.1.1 and 3.1.2 should go in a supplementary document.”
8.2. “The Extraction of the E.coli spectra section with figure 13 could go in the supplementary document.”
8.3. “Sections 4.1 to 4.3 could go in the supplementary document as well.”
The idea is to shorten the paper by moving some parts of the paper in a supplementary data document.
We mostly agree with this but we would like to recall that the goal of this paper is twice. (i) To describe how absorption spectra of bacteria can be modeled so that readers can transpose the method to bacteria they are working with (section 3.1 and 3.2). (ii) To show how this mathematical description can be used for different purposes (section 3.4). This is why:
- Point 8.1.
Sections 3.1.1 and 3.1.2 are fully part of the mathematical modeling of the spectral absorption behavior of bacteria. To our opinion, it must remain in the main text.
Action made: none.
- Point 8.2.
We agree moving, not only figure 13 but also the whole sub-section titled “Extraction of the E. coli spectra” to the supplementary data document. Indeed, this sub-section is a bit aside the two main goals of this paper.
Action made: the supplementary data document has been created and includes this section. Figure numbering has been adapted.
- Point 8.3.
The suggestion is to move remarks concerning the format of the data, the units used for bacteria concentrations and the reason why we express concentration in terms of decimal logarithm. For reason regarding the above mentioned first goal of our paper, we prefer these sections to remain in the discussion section.
Action made: none.
We propose an extra modification in order to shorten the paper. It was not asked by the reviewer but is in line with the suggestions made.
As stated above, section mentioned in point j is a bit aside the main scopes of this paper. We think section 4.4.1 is also a bit aside the goals of the paper. We propose to move it in the supplementary data document.
Action made: this section has been removed and included in the supplementary data document. The text has been updated accordingly.

Round 2
Reviewer 1 Report
The manuscript may be accepted for publication after a minor revision:
1. "Large" and "small" wavelengths are not used in optics at all and it is a nonsense. Please, use "long" and "short" wavelengths, respectively.
2. Although the authors insist on using "absorption" instead of "attenuation" in this paper because it is "more convenient for our purposes", in this study attenuation was measured, not absorption. The reviewer leaves it up to the Editor of this Journal.

Author Response
Second response to reviewer #1, manuscript 2263849
Again, the authors would like to thank the reviewer for the attention paid to our manuscript.
Preliminary remark.
Before to answer to reviewer’s comments and actions made to take remarks into account, we would like to recall that the goal of this paper is to provide biologists and micro-biologists with optical based methods to monitor bacteria growth and to simultaneously monitor concentrations of biological cells in mixtures (description of contaminated cell culture for example).
We acknowledge that the term “attenuation” is more appropriate than the term “absorption” since light-matter interaction processes occurring in the spectroscopy cuvettes implies both absorption and diffusion. However, “attenuation” is not used in biology or micro-biology. We propose the below explained modifications. Both terms are used at the beginning of the manuscript, remarks concerning “absorption” and “attenuation” are added since the beginning of the document and why we use “absorption” in the rest of the document.
Note that if only “attenuation” is used, the paper would not appear in bibliographic researches conducted by colleagues to whom the paper is addressed to and for whom “attenuation” is not a term commonly used.
Reviewer’s comments and tentative answers.
- "Large" and "small" wavelengths are not used in optics at all and it is a nonsense. Please, use "long" and "short" wavelengths, respectively.
We agree “large” and “small” wavelengths are not used in optics.
Action made: corrections have been made.
- Although the authors insist on using "absorption" instead of "attenuation" in this paper because it is "more convenient for our purposes", in this study attenuation was measured, not absorption. The reviewer leaves it up to the Editor of this Journal.
We acknowledge that “more convenient to our purpose” was not adequate.
We also totally agree with the reviewer concerning the fact that attenuation is actually what is measured. However, just as "small" or "large" are not used in optics, "attenuation" is not used in biology and/or micro-biology.
Indeed, everything is a question of definition. Commercial systems used in biology provide spectral information either in “transmission”, “transmittance” or “absorbance”, all of them being related by usual spectroscopy definitions. In fact, the only measurement made with these commercial systems is “transmittance” transformed in “transmission” expressed in percentage or absorbance by a logarithm transformation. Absorption is simply (100-transmission) regardless of the light-matter interaction occurring either in the spectroscopy cuvettes or in the well-plates.
Again, we agree with the reviewer that these commonly accepted definitions in the field of biology are not strictly speaking correct. However, in order to keep this paper bibliographically visible, we propose the following actions (in addition to the specifically written section 4.10 and various mentions in the first revised version).
Action made: we changed “absorption” by “absorption/attenuation” in the title and in sections 1 and 2. In addition to what has already be modified for the first revision, we added a preliminary remark in section 2.5 to explain why the proper term would be “attenuation” and why, despite this, we continue to use the term absorption commonly used by researchers to whom this paper is addressed.
